# Management of Polypharmacy and Potential Drug–Drug Interactions in Patients with Mycobacterial Infection: A 1-Year Experience of a Multidisciplinary Outpatient Clinic

**DOI:** 10.3390/antibiotics12071171

**Published:** 2023-07-10

**Authors:** Dario Cattaneo, Alessandro Torre, Marco Schiuma, Aurora Civati, Samuel Lazzarin, Giuliano Rizzardini, Andrea Gori, Spinello Antinori, Cristina Gervasoni

**Affiliations:** 1Department of Infectious Diseases, ASST Fatebenefratelli-Sacco University Hospital, 20157 Milan, Italy; dario.cattaneo@asst-fbf-sacco.it (D.C.); giuliano.rizzardini@asst-fbf-sacco.it (G.R.); spinello.antinori@unimi.it (S.A.); 2Gestione Ambulatoriale Politerapie (GAP) Outpatient Clinic, ASST Fatebenefratelli-Sacco University Hospital, 20157 Milan, Italy

**Keywords:** mycobacterial infections, polypharmacy, drug–drug interactions, outpatient clinic

## Abstract

In 2022, we opened an outpatient clinic for the management of polypharmacy and potential drug–drug interactions (pDDIs) in patients with mycobacterial infection (called GAP-MyTB). All patients who underwent a GAP-MyTB visit from March 2022 to March 2023 were included in this retrospective analysis. Fifty-two patients were included in the GAP-MyTB database. They were given 10.4 ± 3.7 drugs (2.8 ± 1.0 and 7.8 ± 3.9 were, respectively, antimycobacterial agents and co-medications). Overall, 262 pDDIs were identified and classified as red-flag (2%), orange-flag (72%), or yellow-flag (26%) types. The most frequent actions suggested after the GAP-MyTB assessment were to perform ECG (52%), therapeutic drug monitoring (TDM, 40%), and electrolyte monitoring (33%) among the diagnostic interventions and to reduce/stop proton pump inhibitors (37%), reduce/change statins (14%), and reduce anticholinergic burden (8%) among the pharmacologic interventions. The TDM of rifampicin revealed suboptimal exposure in 39% of patients that resulted in a TDM-guided dose increment (from 645 ± 101 to 793 ± 189 mg/day, *p* < 0.001). The high prevalence of polypharmacy and risk of pDDIs in patients with mycobacterial infection highlights the need for ongoing education on prescribing principles and the optimal management of individual patients. A multidisciplinary approach involving physicians and clinical pharmacologists could help achieve this goal.

## 1. Introduction

Even if major improvements in therapeutic regimens and treatment outcomes have been progressively achieved [1,2,3], mycobacterial infection remains one of the leading causes of death worldwide [4,5]. Important challenges related to the management of antimycobacterial treatments are represented by the issues of polypharmacy and the consequent risks of potential drug–drug interactions (pDDIs) [6,7]. Indeed, rifampicin and isoniazide are, respectively, an inducer and an inhibitor of cytochrome P450 isoenzymes, whereas some of the other antimycobacterial drugs can interfere with the activity of membrane-transport proteins, resulting in many pharmacokinetic-driven pDDIs. Additionally, some antimycobacterial agents could cause some important pharmacodynamic interactions, such as the potential additional effects of delamanid and bedaquiline if combined with other drugs known to cause QT prolongation [8,9]. This scenario may become even more complex by considering that patients with mycobacterial infections, mainly the older ones with comorbidities, may be treated with concomitant medications that can themselves act as perpetrators or victims of pDDIs regardless of the antimycobacterial therapies involved [10,11,12]. An important example is represented by the presence of concomitant infection with HIV. Indeed, Resende et al. have recently shown that more than 95% of their patients co-infected with tuberculosis (TB) and human immunodeficiency virus (HIV) infection were exposed to pDDIs [13]. The most frequent interactions were between anti-TB and antiretroviral drugs, which can cause therapeutic ineffectiveness and major adverse reactions. The authors also reported a significant association between contraindicated and moderate pDDIs with excessive polypharmacy and hospitalization in these patients. Taken together, these findings suggest that the concomitant use of antimycobacterial agents and drugs to treat other diseases can cause pDDIs, eventually leading to unfavorable clinical outcomes. This has been recently confirmed by Noor et al., who reported that 36% of the 436 patients hospitalized with TB experienced adverse effects of anti-TB drugs [14]. Potential adverse outcomes of the most frequent DDIs were hepatotoxicity, decreased drug effectiveness, QT interval prolongation, nephrotoxicity, and gastrointestinal ulceration. For these reasons, in March 2022 we expanded our outpatient clinic for the management of polypharmacy in patients living with HIV (Gestione Ambulatoriale Politerapie (GAP)) and in patients with mycobacterial infections (called GAP-MyTB). The main aims of the GAP clinic are to assess whether the patients are treated with drug combinations which are contraindicated because of known or predictable DDIs, assess the clinical and/or pharmacokinetic relevance of the pDDIs, perform therapeutic drug monitoring (TDM)/pharmacogenetic tests (when deemed appropriate), and provide written advice as to how the treatments should be modified where possible. We also assessed the anti-cholinergic cognitive burden (ACB), a key marker of appropriate medication prescription which refers to the cumulative effect of taking one or more medications with anticholinergic activity [15]. Indeed, it has been shown that the cumulative effect of multiple medicines with anti-cholinergic properties can have an adverse impact on cognition and physical function and increase the risk of falls and mortality, mainly in patients aged >65 years (reviewed in [15]).

The GAP clinic service is not offered to all patients, being an on-demand service. The consultancy requests to the GAP clinic mainly came from colleagues in the infectious diseases department of our hospital. The GAP-MyTB clinic applies the same methodology previously described in people with HIV [16,17]. Here, we describe the results of our 1-year experience. The main goals of the present study were to: (a) characterize the number and types of co-medications given to patients with mycobacterial infections; (b) assess the overall risk of pDDIs (between anti-TB drugs and non anti-TB drugs and between non anti-TB drugs, regardless of the TB therapies involved) and ACB; and (c) describe the potential added value of TDM/pharmacogenetic tests for the optimization of TB treatment in real-life settings.

## 2. Results

### 2.1. Patient Characteristics and TB Treatment

Fifty-two patients were included in the GAP-MyTB database (Table 1). They were mostly men (57.7%), Caucasian (69.2%), and had a mean age of 61 ± 16 years (50% were over 65 years old); 40.4% of them had active TB disease (mainly with pulmonary localization), whereas the remaining had TB infection (25.0%) or infections caused by non-tuberculous mycobacteria (34.6%). Overall, 17 different antimycobacterial regimens were identified in the 52 patients. The most frequent were rifampicin-isoniazid (26.9%), rifabutin-azithromycin-ethambutol (25.0%), rifampicin-isoniazid-ethambutol-pyrazinamide (15.4%), and rifabutin-isoniazid (5.8%).

### 2.2. Co-Medications, ACB Scale, and pDDIs

The patients enrolled in the GAP-MyTB database were given 7.8 ± 3.9 concomitant medications (ranging from 2 to 19 drugs for a total of 372 prescriptions) in addition to their antimycobacterial treatments (2.8 ± 1.0) for a total of 10.4 ± 3.7 drugs. No significant differences in the number of co-medications were found when stratifying the patients according to age (7.2 ± 4.0 versus 8.4 ± 3.9 drugs in patients < 65 years versus >65 years; *p* = 0.265). As highlighted in Figure 1, the most commonly prescribed co-medication classes were dietary supplements (22.0%), antihypertensives (18.0%), central nervous system (CNS) drugs (10.8%), proton pump inhibitors (PPIs, 8.3%), and antiretrovirals (5.6%). In total, 262 pDDIs were identified, which mainly included antimycobacterial drugs (67.9%). No significant differences in the scoring of pDDIs were found when comparing antimycobacterial drugs with co-medications concerning red-flag (1.7% versus 2.4%), orange-flag (73.6% versus 69.0%), or yellow-flag (25.3% versus 27.4%) pDDIs. As shown in Table 2, four out of the five red-flag pDDIs involved a PPI, while two out of five involved the risk of inactivation of clopidogrel. The large majority of orange-flag pDDIs involved rifampicin and its inductive effects on the metabolism of co-medications (i.e., antihypertensives, hypolipidemic agents, glucorticoids, etc.). Examples of yellow-flag pDDIs involved the mild inhibitory effect of isoniazid on the metabolism of co-medications or the isoniazid-induced decreased exposure of pyridoxine.

An ACB score > 3 was observed in 17.3% of patients overall; in 16.0% of patients aged 65 years and over; and in 18.5% of patients aged under 65 years.

### 2.3. Proposed Actions Identified during the GAP-MyTB Visits

Proposed actions have been identified in 50 out of the 52 patients (96.2%) enrolled in the GAP-MyTB database. As shown in Table 3, these actions were divided into diagnostic interventions and changes in the pharmacological therapies. Among the former, an electrocardiogram (ECG) was suggested in 51.9% of patients, mostly related to the associations of drugs that could potentially prolong the QT interval. Additional diagnostic evaluations included: the TDM of both TB drugs (such as rifampicin, levofloxacin, linezolid, etc.) and, in some cases, of co-medications (such as antipsychotics, antiepileptic drugs); frequent electrolyte monitoring (mainly for the risk associated with hypokalemia); and pharmacogenetic tests (N-acetyltransferase 2 (NAT2) for the geno-phenotyping of isoniazid and in two cases for the genotyping of cytochrome 2 C19 (CYP2C19) to facilitate encoding for the key metabolic enzyme involved in the activation of clopidogrel).

Among the pharmacologic interventions, the most frequent action was to stop (or to reduce) the PPI (36.5%), followed by changes in the statin (13.5%, reduction in the dose or change with another statin at lower risk of pDDI). In the four patients aged >65 years with ACB > 3, we suggested a re-evaluation by the psychiatrist (in most cases we suggested to replace quetiapine with another antipsychotic characterized by a lower ACB) to reduce the overall anticholinergic burden. In 5.8% of cases, the suggestion related to changes in timing of daily drugs intake (i.e., to delay the administration bi- or trivalent cations acting as chelants that could potentially reduce the absorption of some drugs).

#### 2.3.1. TDM of Rifampicin

The TDM was available for 26 out of the 30 TB patients treated with rifampicin (86.7%). Plasma peak concentrations (Cmax, 10.4 ± 4.9 mg/L) were recorded, respectively, at 2 h (85.7%) and 4 h (14.3%) after the morning administration of oral rifampicin. Sub-therapeutic rifampicin Cmax concentrations (<8 mg/L) were found in 39% of TB patients; no patients had rifampicin Cmax concentrations > 24 mg/L (Table 4). The observed rifampicin underexposure resulted in a TDM-guided daily dose increment from 645 ± 101 mg (before TDM) to 793 ± 189 mg/day (after TDM, *p* < 0.001) starting at 11.0 ± 11.3 days after the blood sampling. At the next TDM after dose adjustment, all but one patients reached therapeutic rifampicin Cmax concentrations (13.6 ± 3.8 mg/L).

#### 2.3.2. NAT2 Phenotyping

As shown in Table 4, the majority of patients resulted as slow (48.2%) or intermediate (44.4%) NAT2 acetylators. In the only two patients (7.4%) that were rapid acetylators, the dose of isoniazid was increased to compensate for the fast drug metabolization. Conversely, 3 out of the 29 patients on isoniazid (10.3%) required a dose reduction for the development of hepatic toxicity attributed to isoniazid (all three patients were slow NAT2 acetylators).

## 3. Discussion

In this cross-sectional study, we reported the 1-year experience of GAP-MyTB, an outpatient clinic specifically developed in our hospital for the optimization of complex therapies in patients with mycobacterial infections, as previously performed in the field of HIV [16,17]. A high rate of polypharmacy was observed, with a mean of 8 drugs concomitantly prescribed in addition to the antimycobacterial regimens. As expected, the observed heavy polypharmacy resulted in more than 260 pDDIs recorded. Remarkably, this is likely to be an underestimated value considering our decision to group pDDIs involving two or more drugs acting as perpetrators for the same co-medication instead of considering each paired drug interaction separately, as explained in the Section 4.

The observed pDDIs were mainly driven by antimycobacterial drugs (around 70%), with three red-flag DDIs. The first one involved the concomitant administration of azithromycin and lithium increasing the risk of cardiotoxicity for the added effect of both drugs on QT interval prolongation. The second one involved isoniazid which decreased the conversion of clopidogrel to its active form, ultimately reducing the response to the antiplatelet agent [18]. The last red-flag DDI was related to the risk of sub-therapeutic voriconazole exposure and high risk of failure to therapy for the concomitant administration of rifabutin (moderate inducer of CYP3A4) [19]. This severe pDDI was handled per clinical practice by adjusting the dose of voriconazole based on TDM results and by reducing the dose of rifabutin [20].

Remarkably, four out of the five red-flag DDIs involved PPIs. These drugs can, in fact: (a) have addictive effects when combined with other drugs that can cause QT prolongation (such as azithromycin and lithium); (b) inhibit the activation of clopidogrel to its active form via inhibition of CYP2C19 (especially in patients with a genetically-based impairment of enzyme activity and/or when combined with inhibitory agents such as isoniazid); and (c) increase the exposure and toxicity of methotrexate by decreasing the renal clearance. Remarkably, 31 out of the 52 patients from the GAP-MyTB database were receiving chronic treatment with PPIs, the large majority having no therapeutic indication (61%) for their use. Accordingly, the most frequent pharmacologic intervention from the GAP-MyTB clinics was to stop/reduce the PPI. Our data reinforce the concept, already observed during COVID-19, that despite information campaigns (either within the hospital or in the community) and other efforts to promote the correct use of PPIs, overuse remains excessive and their potential for causing clinically relevant DDIs is still underestimated [21,22]. Besides their potential to cause DDIs, patients taking PPIs for a long time are at risk for worse clinical outcomes, including increased risk of kidney, liver, and cardiovascular disease, dementia, enteroendocrine tumors of the gastrointestinal tract, susceptibility to respiratory and gastrointestinal infections, and impaired absorption of nutrients [23,24,25].

Sixteen percent of the patients from the GAP-MyTB clinics aged > 65 years (50%) had an ACB of ≥ 3, a score which has been associated with worse outcomes (i.e., cognitive impairment, falls) [16,17]. These data suggest an ongoing suboptimal physicians’ perceptions of the risks associated with the cumulative effect of anti-cholinergic drugs [26,27]. Some of the available free web-based drug interaction checkers can easily assess the anticholinergic burden using different scales. All physicians should ideally use these tools in their clinical practice every time they treat patients aged >65 years (regardless of the presence of mycobacterial infections or other infectious diseases).

Rifampicin was the most frequently used drug in our TB patients. The adequacy of its dosage should therefore be mandatorily verified. Indeed, recent literature has documented that a significant proportion of TB patients treated with the conventional 600 mg once-daily dose may be exposed to sub-therapeutic rifampicin concentrations [28,29]. This trend has also been confirmed by our findings. Indeed, nearly 40% of the patients from our database had rifampicin Cmax concentrations below the minimum therapeutic threshold (set at 8 mg/L); this was easily resolved by a TDM-guided 40% increment in the daily drug dose.

More than 90% of the TB patients being treated with isoniazid were genotyped for NAT2 with the goal of characterizing their acetylator phenotype. Indeed, extensive evidence is available showing that NAT2 genotype is one of the most important covariates influencing the plasma concentration of isoniazid [30]. Among the three NAT2 acetylator phenotypes, rapid acetylators achieve the lowest and slow acetylators achieve the highest plasma concentration of isoniazid. Moreover, NAT2 slow acetylator TB patients have been reported to have a comparatively higher early bactericidal activity of isoniazid than rapid acetylators [31]. For this reason, the dose of isoniazid was increased in the two patients from our database with the rapid NAT2 acetylator phenotype.

Previous studies have dealt with the issue of pDDIs and polypharmacy in TB patients [11,12,13,14]. All these studies documented that patients with TB present with a considerable number of clinically important pDDIs, especially those hospitalized and those with excessive polypharmacy. We believe that the novelty of our study could rely on the thorough assessment of the pharmacologic burden of our patients, focusing not only on the assessment of pDDIs between anti-TB drugs and non-anti-TB drugs but also on assessing the pDDIs between non-anti-TB drugs and on routinely performing TDM and PG analyses when deemed appropriate. Last but not least, we systematically described the diagnostic and pharmacologic interventions provided by the physicians involved in the management of TB patients.

Potential limitations of the present investigation are: the retrospective design which may have introduced bias and confounding factors that were not accounted for; the single-center study which may limit the generalizability of these findings to other settings; the small sample size of the study which may limit the statistical power of the analysis; and the lack of data on the long-term outcomes of the interventions suggested by the GAP-MyTB clinic, which may be important to assess the effectiveness of the approach. Nevertheless, we are confident that our study could be instrumental in underlining the importance of a multidisciplinary approach (i.e., the involvement of infectious disease physicians, clinical pharmacologists, pharmacists, etc.) to recognize the high prevalence of polypharmacy and risk of pDDIs in patients with mycobacterial infection and in highlighting the need for ongoing education on prescribing principles and the optimal management of individual patients.

## 4. Materials and Methods

### 4.1. Patient Selection and Study Design

Demographic characteristics, information on the type and localization of mycobacterial infections, antimycobacterial agents, and number/class of co-medications were collected in the patients included in the GAP-MyTB database from March 2022 to March 2023. The overall risk of pDDIs between all administered drugs was assessed using INTERcheck WEB (https://intercheckweb.marionegri.it, accessed on 15 April 2023) and Medscape Drug Interaction checker (https://reference.medscape.com/drug-interactionchecker, accessed on 15 April 2023); HIV Drug Interactions checker (https://www.hiv-druginteractions.org/checker. accessed on 15 April 2023) was used for patients co-infected with mycobacterial infections and HIV. pDDIs were classified as red-, orange-, or yellow-flag types based on their severity and clinical relevance, as follows. Red-flag: drug combinations that should be avoided; orange-flag: drug combinations that may require close monitoring and/or drug dose adjustments to avoid potentially serious clinical consequences; yellow-flag: drug combinations with minor clinical relevance. The first assessment/scoring of the severity of each DDI was performed independently by DC and subsequently reassessed by CG based on their extensive experience on pDDIs developed with the GAP-HIV clinics [16,17]. The concomitant presence of two or more drugs acting together as perpetrator was counted as a single pDDI instead of considering each paired interaction separately. For instance, isoniazid+PPI+clopidogrel was counted as one DDI and not as two DDIs (one pDDI between isoniazid+clopidogrel and one pDDI between PPI and clopidogrel) since both isoniazid and PPI reduced the bioactivation of clopidogrel. Some pDDIs, although scored as red- or orange-flag types by some drug interaction checkers, were not judged as clinically relevant by the investigators if they could be easily handled in day-by-day practice, as in the case of calcium channel blockers or alpha 1 reductase inhibitors and ritonavir/cobicistat [32,33].

The burden of medications with anticholinergic effects was also estimated using the ACB scale [15]. An ACB score ≥ 3 has been associated with a significantly increased risk of adverse events, including cognitive impairment, falls, and blurred vision in patients aged >65 years.

A written report summarizing the pDDIs based on clinical relevance, the additional diagnostic interventions (i.e., TDM of rifampicin and/or NAT2 genotyping to assess the rate of isoniazid acetylation), as well as practical suggestions as to how the pharmacologic treatments should be modified, was provided for the attending physicians at the end of the GAP-MyTB visit.

The study was approved by our Ethics Committee (Comitato Etico Interaziendale Area 1, Milan, Italy (Protocol No. 11903)). All patients gave written informed consent for medical procedures/interventions performed for routine treatment purposes, according to the Ethics Committee (Comitato Etico Interaziendale Area 1, Milan, Italy).

### 4.2. Assessment of Rifampicin Plasma Concentrations and TDM

Rifampicin plasma concentrations were measured by a liquid chromatography method coupled with tandem mass spectrometry (LC–MS/MS) [34]. After precipitation of plasma proteins with an organic solvent, the quantification of rifampicin was carried out by electrospray positive ionization mass spectrometry in the multiple-reaction monitoring mode. The method was linear over the concentration ranges of 0.2–100 mg/L; between- and within-day imprecision and inaccuracy were less than 15% during each analytical run. The performance of the method was tested during each analytical run using internal quality controls.

The TDM of rifampicin relies on the assessment of Cmax, with a therapeutic range of 8–24 mg/L [35]. Considering the wide variability of rifampicin absorption after oral drug administration, blood samples for the assessment of Cmax were collected at 2, 4, and 6 h after the fasting morning oral administration (Cmax was defined as the highest concentrations measured during this sampling interval). Additionally, the rifampicin area under the curve from 0 to 24 h (AUC0–24) was estimated using the equation proposed by Magis-Escurma et al. [36].

### 4.3. Pharmacogenetic Tests

Genomic DNA was isolated from peripheral blood cells using an automated DNA extraction system (EZ1 Advanced XL, Qiagen, Hilden, Germany). DNA concentration and purity were evaluated by absorbance methodology using a NanoDrop 1000 Spectrophotometer V3.7 (Thermo Fisher Scientific, Waltham, MA, USA). The following allelic variants were investigated: NAT2*5 c.341 T > C (rs1801280), NAT2*6 c.590 G > A (rs1799930), NAT2*7 c.857 G > A (rs1799931), and NAT2*14 c.191 G > A (rs1801279). All genotypes were determined by real-time PCR using the LightSNiP (TIB-MolBiol, Berlin, Germany) on a LightCycler 480 (Roche, Basel, Switzerland). Genotyping performance was estimated through the use, in each analysis, of known-genotype internal quality controls.

Based on the presence of the NAT2 allelic variants, the patients were grouped into three acetylator phenotypes (fast, intermediate, and slow) following the PharmGKB classification (available at https://www.pharmgkb.org/vip/PA166170337, accessed on 15 April 2023).

All patients provided additional written signed consent for the pharmacogenetic analyses.

### 4.4. Statistical Analyses

The frequency distribution data are expressed as absolute numbers and percentages, and all of the other measures are expressed as mean values ± standard deviation. Differences between groups were tested using Student’s *t*-test for continuous variables and Pearson’s chi-squared test for dichotomous and unordered categorical data.

## 5. Conclusions

The high prevalence of polypharmacy and risk of pDDIs in patients with mycobacterial infections highlights the need for ongoing education on prescribing principles and the optimal management of individual patients. A multidisciplinary approach involving physicians and clinical pharmacologists could help achieve this goal.

## Figures and Tables

**Figure 1 antibiotics-12-01171-f001:**
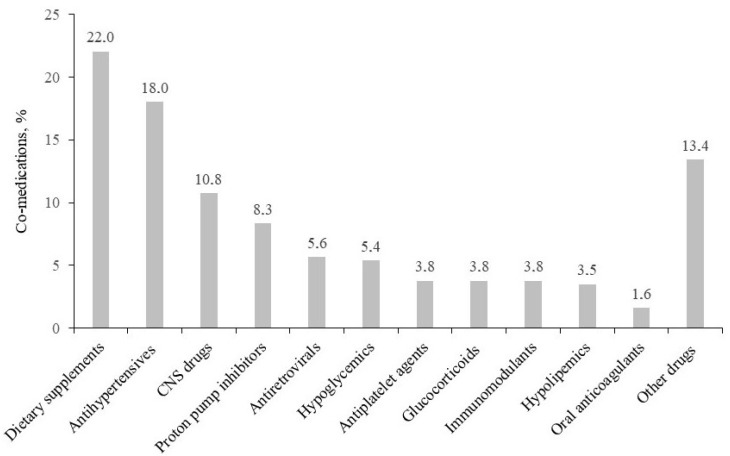
Distribution of the main drug classes of co-medications in the 52 patients with mycobacterial infections included in the GAP-MyTB database (data are given as percentages of the total of non-mycobacterial prescriptions).

**Table 1 antibiotics-12-01171-t001:** Clinical features of the patients included in the database of the GAP-MyTB clinic. Data in parenthesis refer to percentages.

Characteristics	Clinical Features
Patients, *n*	52
Females, *n* (%)	22 (42.3%)
Mean age, years	61 ± 16
HIV co-infection, *n* (%)	8 (15.4%)
Ethnicity (%)	Caucasian (69%), Asian (10%), Hispanic (8%), Black (8%), Arab (6%)
TB disease, *n* (%)	21 (40.4%)
Localization (*n*)	Pulmonary (*n* = 11), ocular (*n* = 4), abdominal (*n* = 2), cerebral (*n* = 1), renal (*n* = 1), pulmonary/abdominal (*n* = 1), pulmonary/cerebral (*n* = 1)
TB infection, *n* (%)	13 (25.0%)
NTM, *n* (%)	18 (34.6%)
Anti-tubercular treatments (*n*)	Rifampicin (*n* = 30), ethambutol (*n* = 29), isoniazid (*n* = 29), rifabutin (*n* = 17), azithromycin (*n* = 16), pyrazinamide (*n* = 12), linezolid (*n* = 2), amikacin (*n* = 2), levofloxacin (*n* = 2),clarithromycin (*n* = 1), bedaquiline (*n* = 1), moxifloxacin (*n* = 1), clofazimine (*n* = 1)

TB: tuberculosis; NTM: non-tuberculous mycobacteria.

**Table 2 antibiotics-12-01171-t002:** Total number of drugs, potential drug–drug interactions (pDDIs), and anticholinergic burden recorded in the database of the GAP-MyTB clinic.

Type of pDDI	Overall	TB Drugs	Co-Medications
Drugs, *n*	10.4 ± 3.7	2.8 ± 1.0	7.8 ± 3.9
pDDIs	262	178	84
Red-flag pDDIs	5	3 Azithromycin/lithium/omeprazole Isoniazid/clopidogrel/rabeprazole Voriconazole//rifabutine	2 Methotrexate/omeprazole Clopidogrel/rabeprazole
Orange-flag pDDIs	189	130	59
Yellow-flag pDDIs	68	45	23
ACB ≥ 3	9 (17.3%)	5 patients ≤ 65 years and 4 patients > 65 years	

pDDI: potential drug–drug interaction; ACB: anticholinergic cognitive burden scale; red-flag: drug combinations that should be avoided; orange-flag: drug combinations that may require close monitoring and/or drug dose adjustments to avoid potentially serious clinical consequences; yellow-flag: drug combinations with minor clinical relevance.

**Table 3 antibiotics-12-01171-t003:** Interventions suggested at the end of the GAP-MyTB visits (in addition to the routine care).

Diagnostic Intervention	Frequency, *n* (%)
Perform electrocardiogram	27 (51.9%)
Perform therapeutic drug monitoring	21 (40.4%)
Monitor serum electrolytes	17 (32.7%)
Perform pharmacogenetic test	11 (21.2%)
Monitor metabolic assessment	8 (15.4%)
Monitor blood pressure	8 (15.4%)
Monitor liver function	7 (13.5%)
Monitor renal function	2 (3.8%)
Monitor thyroid hormones	2 (3.8%)
Monitor respiratory functionality	2 (3.8%)
**Changes in pharmacologic therapies**	**Frequency, *n* (%)**
Reduce/stop proton pump inhibitor	19 (36.5%)
Reduce/change statin	7 (13.5%)
Reduce anticholinergic burden	4 (7.7%)
Change timing of daily drug intake	3 (5.8%)
Reduce/stop benzodiazepine	2 (3.8%)
Change antiplatelet	2 (3.8%)
Change bisphosphonate	1 (1.9%)
Change oral anticoagulant	1 (1.9%)
Stop diuretic	1 (1.9%)
**Patients with no suggestions**	**2 (3.8%)**

**Table 4 antibiotics-12-01171-t004:** Therapeutic drug monitoring and geno-phenotypic analyses of TB drugs.

Diagnostic Intervention	Data
TDM of rifampicin, *n* (% of treated patients)	26 (86.7%)
Mean dose of rifampicin before TDM, mg/day	645 ± 101
[Rifampicin] C2, mg/L	9.9 ± 5.5
[Rifampicin] C4, mg/L	8.1 ± 2.8
[Rifampicin] C6, mg/L	5.6 ± 3.1
[Rifampicin] Cmax, mg/L	10.4 ± 4.9
Rifampicin AUC0–24, mg×h/L	65.6 ± 31.0
[Rifampicin] Cmax < 8 mg/L, %	38.5%
[Rifampicin] Cmax > 24 mg/L, %	0%
Mean dose of rifampicin after TDM, mg/day	793 ± 189 (+38%) *
Time to rifampicin dose change, days	11.0 ± 11.3
Geno-phenotyping of NAT2, *n* (%) -Rapid NAT2 acetylators -Intermediate NAT2 acetylators -Slow NAT2 acetylators	27 (93.1%) 2 (7.4%) 12 (44.4%) 13 (48.2%)
Isoniazid dose modifications, *n* (%) -Dose reduction -Dose increase	5 (18.5%) 3 (toxicity) 2 (rapid acetylators)

TB: tuberculosis; TDM: therapeutic drug monitoring; C2, C4, C6: concentrations collected at 2, 4, and 6 h after drug administration, respectively; Cmax: peak concentration; AUC: area under the curve; NAT2: N-acetyl transferase 2. * *p* < 0.001 versus dose before TDM.

## Data Availability

Data are available from the authors upon reasonable request.

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
