# Peer review of "Management of Polypharmacy and Potential Drug–Drug Interactions in Patients with Mycobacterial Infection: A 1-Year Experience of a Multidisciplinary Outpatient Clinic"

_antibiotics, 2023, doi:10.3390/antibiotics12071171_

Round 1

Reviewer 1 Report

I read, Management of polypharmacy and potential drug-drug interactions in patients with Mycobacterial infection: A 1-year experience of a multidisciplinary outpatient clinic, with interest. In this manuscript, the authors were aimed to describe the results of 1-year experience. The findings of this paper can be used to improve prescribing practices and optimize the management of patients with Mycobacterial infection.

 I have some questions and suggestions.

1. Can you explain why this study is new or telling new things?

2. Introduction: Please give specific objective in this study

3. What is the purpose of GAP-MyTB clinic and what patient population does it serve?

4. How many patients were included in this retrospective analysis, and what was the average number of drugs they were taking?

5. What are red-flag, orange-flag, and yellow-flag pDDIs? Can you provide examples for each category?

6. Which diagnostic interventions were most commonly recommended after a GAP-MyTB assessment?

7. How did TDM-guided dose increment improve rifampicin exposure in patients with Mycobacterial infection?

8. Discussion: Please compare the results of this study with the results of other studies for a more in-depth discussion.

9. The limitations of this paper are not explicitly mentioned in the given information. However, some possible limitations could be:

•            The study was conducted in a single center, which may limit the generalizability of the findings to other settings.

•            The retrospective design of the study may have introduced bias and confounding factors that were not accounted for.

•            The sample size of the study was relatively small, which may limit the statistical power of the analysis.

•            The study did not evaluate the long-term outcomes of the interventions suggested by the GAP-MyTB clinic, which may be important to assess the effectiveness of the approach.

10. Please provide more data on the importance of physicians and pharmacy around the world to recognize the high prevalence of polypharmacy and risk of pDDIs in patients with Mycobacterial infections highlights the need for ongoing education on prescribing principles and the optimal management of individual patients.

11. Lines 346-349 are duplicates of line 352-355. please rewrite

Minor editing of English language required

Author Response

  1. Can you explain why this study is new or telling new things?

In the revised version of the discussion we have explained what, in our mind, our study could add to already available literature in the field. In particular, we believe that the novelty of our study could rely on the thorough assessment of the pharmacologic burden of our patients, focusing not only on the assessment of pDDIs between anti-TB drugs and non anti-TB drug but also assessing the pDDIs between non anti-TB drugs, and by routinely perform TDM and PG analyses when deemed appropriate. Last but not the least, we systematically described the diagnostic and pharmacologic interventions provided to the physicians involved in the management of TB patients (lines 280-289 of the revised manuscript). 

  1. Introduction: Please give specific objective in this study

A more specific description of the main objectives of the present study has been added in the revised introduction. The main goals of the present study are to A) characterize the number and types of co-medications given to patients with Mycobacterial infection; B) assess the overall risk of pDDIs (between anti-TB drugs and non anti-TB drugs and between non anti-TB drugs regardless of TB therapies) and ACB; C) describe the potential added value of TDM/pharmacogenetic tests for the optimization of TB treatment in real-life settings (lines 76-81 of the revised manuscript).

  1. What is the purpose of GAP-MyTB clinic and what patient population does it serve?

The GAP-MyTB clinic service is not offered to all patients but is an on-demand activity. The consultancy requests to the GAP-MyTB clinic mainly came from colleagues in the Infectious Diseases Department of our hospital (lines 72-74 of the revised manuscript).

  1. How many patients were included in this retrospective analysis, and what was the average number of drugs they were taking?

As already mentioned in the first submission, 52 patients were included in the GAP-MyTB database, and they were given 10.4±3.7 drugs.

  1. What are red-flag, orange-flag, and yellow-flag pDDIs? Can you provide examples for each category?

As requested by the Reviewer, some examples of orange and yellow-flag pDDIs were added in the revised manuscript (lines 112-116 of the revised manuscript).

  1. Which diagnostic interventions were most commonly recommended after a GAP-MyTB assessment?

The most commonly recommended diagnostic interventions were ECG, TDM of both TB-drugs and co-medications, frequent electrolyte monitoring and pharmacogenetic tests (see Table 3 for detailed information on the frequency of the recommendations).

  1. How did TDM-guided dose increment improve rifampicin exposure in patients with Mycobacterial infection?

We apologize for the missing information. As better clarified in the revised manuscript, all but one patients with suboptimal exposure to rifampicin at the first visit, had rifampicin therapeutic Cmax concentrations at the second TDM assessment (lines 183-184 of the revised manuscript).

  1. Discussion: Please compare the results of this study with the results of other studies for a more in-depth discussion.

A more balance discussion has been provided in the revised manuscript, by comparing our results with those from previously published studies in the field (lines 280-289 of the revised manuscript).

  1. The limitations of this paper are not explicitly mentioned in the given information. However, some possible limitations could be:
  • The study was conducted in a single center, which may limit the generalizability of the findings to other settings.
  • The retrospective design of the study may have introduced bias and confounding factors that were not accounted for.
  • The sample size of the study was relatively small, which may limit the statistical power of the analysis.
  • The study did not evaluate the long-term outcomes of the interventions suggested by the GAP-MyTB clinic, which may be important to assess the effectiveness of the approach.

We thank the Reviewer for the critical assessment of our study. We agree with all the potential limitations identified by the Reviewer, which have been clearly underlined in the revised discussion (lines 290-295 of the revised manuscript).

  1. Please provide more data on the importance of physicians and pharmacy around the world to recognize the high prevalence of polypharmacy and risk of pDDIs in patients with Mycobacterial infections highlights the need for ongoing education on prescribing principles and the optimal management of individual patients.

The Reviewer has raised very important points. These concepts have been added and discussed in the revised version of the manuscript. In particular, at the end of the discussion we have added the following concept “Nevertheless, we are confident that our study, could be instrumental in underlying the importance of a multidisciplinary approach (i.e. infectious diseases physicians, clinical pharmacologists, pharmacists, etc.) to recognize the high prevalence of polypharmacy and risk of pDDIs in patients with Mycobacterial infections highlights the need for ongoing education on prescribing principles and the optimal management of individual patients (lines 296-301 of the revised manuscript).

  1. Lines 346-349 are duplicates of line 352-355. please rewrite

We apologize for the duplication that has been removed in the revised manuscript.

Reviewer 2 Report

The authors describe their experience in a one-year outpatient service for the management of polypharmacy and potential drug-drug interactions (pDDI) in patients with mycobacterial infection (referred to as GAP-MyTB). The results were obtained from a retrospective study with all patients attending the GAP-MyTB service from 03/2022 to 03/2023, which included fifty-two patients (registered in the internal database). According to their analysis, 10.4±3.7 drugs (on average) were administered to patients in different regimens, of which 2.8±1.0 were antimycobacterial agents and 7.8±3.9 other drugs. A total of 262 pDDI were identified and classified as red (2%), orange (72%), or yellow (26%) flags. Finally, they concluded that polypharmacy increased the risk of pDDIs in patients with mycobacterial infections and highlighted the need for continued training on prescribing principles and optimal patient management, and that a multidisciplinary approach involving physicians and pharmacologists is highly desirable.

Although the findings of this study impact the field, the following concerns weaken the study and should be adequately addressed before resubmitting this manuscript.

1. Introduction. The background of this study is limited and has not been discussed thoroughly. Therefore, a comprehensive review is recommended, highlighting the hypothesis/question/goal, the uniqueness of the study, and its contributions to the field.

2. The research methodology is standard and straightforward, with limited contributions to the field development. Please make a proper link between the methods and the hypothesis/question/goal of the study, considering the basic standards of scientific research.

3. The effect of polypharmacy on the success of drug treatment in patients with bacterial infections has already been demonstrated. What is the singularity of this study? What are the novel contributions of this study in the field of antibiotics? Please discuss this in greater detail.

A thorough English proofreading is recommended, preferably by a service that employs native speakers with a scientific background.

Author Response

  1. Introduction. The background of this study is limited and has not been discussed thoroughly. Therefore, a comprehensive review is recommended, highlighting the hypothesis/question/goal, the uniqueness of the study, and its contributions to the field.

A more specific description of the main objectives of the present study and GAP-MyTB clinics have been added in the revised introduction (lines 76-81). We believe that the novelty of our study could rely on the thorough assessment of the pharmacologic burden of our patients, focusing not only on the assessment of pDDIs between anti-TB drugs and non anti-TB drugs but also assessing the pDDIs between non anti-TB drugs, and by routinely perform TDM and PG analyses when deemed appropriate. Last but not the least, we systematically described the diagnostic and pharmacologic interventions provided to the physicians involved in the management of TB patients. 

  1. The research methodology is standard and straightforward, with limited contributions to the field development. Please make a proper link between the methods and the hypothesis/question/goal of the study, considering the basic standards of scientific research.

We apologize for the poor quality of the search methodology. The Reviewer should, however, consider that the present study is not a prospective, randomized clinical trial, but a retrospective analysis of an outpatient service that has been implemented by our hospital with the goal to improve the appropriateness of medicine prescriptions in the routine management of TB patients.

  1. The effect of polypharmacy on the success of drug treatment in patients with bacterial infections has already been demonstrated. What is the singularity of this study? What are the novel contributions of this study in the field of antibiotics? Please discuss this in greater detail.

A more balance discussion has been provided in the revised manuscript, by comparing our results with those from previously published studies in the field. In particular, we are confident that our study could be instrumental in underlying the importance of a multidisciplinary approach (i.e. physicians, pharmacists, etc.) to recognize the high prevalence of polypharmacy and risk of pDDIs in patients with Mycobacterial infection highlights the need for ongoing education on prescribing principles and the optimal management of individual patients (280-289, 296-301).

Reviewer 3 Report

The article entitled "Management of polypharmacy and potential drug-drug interactions in patients with Mycobacterial infection: A 1-year experience of a multidisciplinary outpatient clinic"by Cattaneo et al describes the drug-drug interaction and its outcome on treating the disease. The authors used an adequate number of patients. Overall this is an interesting study and opens further thoughts for the disease regimen to treat TB disease.

I do not have major concerns with the submitted manuscript. Only the minor point is the use of abbreviations. The authors used the abbreviations without giving any background information. For example: QT prolongation, PPI etc. 

In sub-heading 2.2, the authors wrote ABC, but in the text, they wrote ACB. Are they similar terms or different, or is it a typing error? If these are different terms, please explain them first. Explain why ACB analysis was performed.

If figure-1 is explained before table-2 then put the figure-1 before table-2 in the manuscript.

The line number 95 is overlapping on the table-1. Please correct.

Satisfactory writing skill. 

Author Response

  1. I do not have major concerns with the submitted manuscript. Only the minor point is the use of abbreviations. The authors used the abbreviations without giving any background information. For example: QT prolongation, PPI etc. 

We thank the Reviewer for the positive comments and apologize for the extensive and inappropriate use of abbreviations. The inconsistencies have been solved in the revised version of the manuscript (with the only exception of QT which is not an abbreviation).

  1. In sub-heading 2.2, the authors wrote ABC, but in the text, they wrote ACB. Are they similar terms or different, or is it a typing error? If these are different terms, please explain them first. Explain why ACB analysis was performed.

ABC was a typing error, sorry for that. More information on the ACB analysis were given in the revised manuscript. We have decided to perform also ACB analysis because the anti-cholinergic burden is an important key marker of appropriate medication prescription. Indeed, it has been shown that the cumulative effect of multiple medicines with anti-cholinergic properties can have an adverse impact on cognition and physical function and increase the risk of falls and mortality mainly, but not exclusively, in patients with >65 years (50% of our patients had >65 years). These concepts have been added in the revised version of the manuscript (lines 65-71 of the revised manuscript).

  1. If figure-1 is explained before table-2 then put the figure-1 before table-2 in the manuscript.

The line number 95 is overlapping on the table-1. Please correct.

The order of Tables and Figures has been revised following the suggestions of the Reviewer.

Reviewer 4 Report

In the present article, Dario Cattaneo et. al; have presented a one-year review of the experience of a multidisciplinary outpatient clinic on the management of polypharmacy and potential drug-drug interaction in patients with mycobacterial infection. There are several points that I would like to mention here for the betterment of the article:

1.     Provide the complete form of the abbreviation when used for the first time in the manuscript.

2.     Please cite some recent references on antimycobacterial drugs in the first paragraph of the Introduction section. For example, Conradie, F. et al. Treatment of highly drug-resistant pulmonary tuberculosis. N. Engl. J. Med. 382, 893–902 (2020), doi.org/10.1080/1061186X.2018.1473407;

3.     In Table 1, please use a different description rather than saying its data. It’s not clear what the number in the bracket represents. The table legend should be more descriptive and informative.

4.     In Table 2, there are types of pDD1s mentioned but in the text, I couldn’t find what these types represent. And the drugs named in the table should be discussed enough in the discussion section with references.

Author Response

  1. Provide the complete form of the abbreviation when used for the first time in the manuscript.

We apologize for the extensive and inappropriate use of abbreviations. The inconsistencies have been solved in the revised version of the manuscript.

  1. Please cite some recent references on antimycobacterial drugs in the first paragraph of the Introduction section. For example, Conradie, F. et al. Treatment of highly drug-resistant pulmonary tuberculosis. N. Engl. J. Med. 382, 893–902 (2020), doi.org/10.1080/1061186X.2018.1473407.

New references have been added in the revised manuscript (new references #1-3).

  1. In Table 1, please use a different description rather than saying its data. It’s not clear what the number in the bracket represents. The table legend should be more descriptive and informative.

Table 1 has been revised following the suggestions of the Reviewer.

  1. In Table 2, there are types of pDD1s mentioned but in the text, I couldn’t find what these types represent. And the drugs named in the table should be discussed enough in the discussion section with references.

The legend of Table 2 has been revised following the suggestions of the Reviewer (see also lines 227-236 of the revised manuscript). The drugs named in the Table have now been discussed in the revised version of the manuscript (lines 313-316 of the revised manuscript).

Round 2

Reviewer 2 Report

The authors adequately addressed the suggestions and comments. Therefore, the manuscript is approved for publication in ANTIBIOTICS.

Minor editing of English language required.